# Dimethylcysteine (DiCys)/*o*-Phthalaldehyde Derivatization for Chiral Metabolite Analyses: Cross-Comparison of Six Chiral Thiols

**DOI:** 10.3390/molecules26247416

**Published:** 2021-12-07

**Authors:** Ankhbayar Lkhagva, Hwan-Ching Tai

**Affiliations:** 1Department of Chemistry, National University of Mongolia, Ulaanbaatar 14200, Mongolia; a_lkhagva@uncg.edu; 2School of Public Health, Xiamen University, Xiamen 361102, China

**Keywords:** chiral metabolomics, rice water, d-amino acids, enantiomer separation, dimethyl labeling

## Abstract

Metabolomics profiling using liquid chromatography-mass spectrometry (LC-MS) has become an important tool in biomedical research. However, resolving enantiomers still represents a significant challenge in the metabolomics study of complex samples. Here, we introduced *N,N*-dimethyl-l-cysteine (dimethylcysteine, DiCys), a chiral thiol, for the *o*-phthalaldehyde (OPA) derivatization of enantiomeric amine metabolites. We took interest in DiCys because of its potential for multiplex isotope-tagged quantification. Here, we characterized the usefulness of DiCys in reversed-phase LC-MS analyses of chiral metabolites, compared against five commonly used chiral thiols: *N*-acetyl-l-cysteine (NAC); *N*-acetyl-d-penicillamine (NAP); isobutyryl-l-cysteine (IBLC); *N*-(*tert*-butoxycarbonyl)-l-cysteine methyl ester (NBC); and *N*-(*tert*-butylthiocarbamoyl)-l-cysteine ethyl ester (BTCC). DiCys and IBLC showed the best overall performance in terms of chiral separation, fluorescence intensity, and ionization efficiency. For chiral separation of amino acids, DiCys/OPA also outperformed Marfey’s reagents: 1-fluoro-2-4-dinitrophenyl-5-l-valine amide (FDVA) and 1-fluoro-2-4-dinitrophenyl-5-l-alanine amide (FDAA). As proof of principle, we compared DiCys and IBLC for detecting chiral metabolites in aqueous extracts of rice. By LC–MS analyses, both methods detected twenty proteinogenic l-amino acids and seven d-amino acids (Ala, Arg, Lys, Phe, Ser, Tyr, and Val), but DiCys showed better analyte separation. We conclude that DiCys/OPA is an excellent amine-derivatization method for enantiomeric metabolite detection in LC-MS analyses.

## 1. Introduction

In the post-genomics era, metabolomics profiling has become an important tool in biomedical research [1,2,3,4]. For highly complex metabolomes, reversed-phase liquid chromatography–tandem MS (RP-LC-MS/MS) analyses is the standard tool for high-throughput discovery [5,6,7,8,9]. One of the fundamental limitations of RP-LC-MS is the lack of stereoselectivity, but many important metabolites are chiral molecules. Recently, chiral metabolomics has become an area of emerging interest [10,11,12,13,14,15].

Initial interests in chiral metabolomics began with d-amino acids, which are physiologically active substances in mammals [16,17]. In fact, d-serine, d-aspartate, d-alanine, and d-cysteine are found in many tissues and body fluids, and several d-amino acids are important neurotransmitters in the brain [18,19]. Enantiomeric amino acids and their derivatives may be useful biomarkers and novel drug candidates; their detection is important in pharmacological research, clinical analysis, agriculture, and food science [20,21,22]. Using isotope tagging, advanced MS instrumentation, and new MS data analysis schemes, it is possible to carry out non-targeted chiral metabolomics profiling and discover novel chiral biomarkers beyond just amino acids [23].

A classic reagent for the derivatization of amine metabolites is *o*-phthalaldehyde (OPA), widely utilized in commercial amino acid analyzers [24,25,26,27]. The chemical reaction with OPA to form fluorescent isoindole derivatives requires a nucleophilic thiol. Coupling OPA to chiral thiols enables chiral separation via diastereomer formation. Chiral thiols tested for OPA derivatization included *N*-acetyl-l-cysteine (NAC) [28]; *N*-acetyl-d-penicillamine (NAP) [29]; isobutyryl-l-cysteine (IBLC) [30]; *N*-(*tert*-butoxycarbonyl)-l-cysteine methyl ester (NBC) [31]; *N*-(*tert*-butylthiocarbamoyl)-l-cysteine ethyl ester (BTCC) [32]; *N*-*R*-mandelyl-l-cysteine (NMC) [33,34]; and 2,3,4,6-tetra-*o*-acetyl-1-thio-*β*-d-glucopyranose (TATG) [34].

In advanced chiral metabolomics profiling, labeling with heavy isotopes is very important for quantification. To our knowledge, no study has introduced isotope labels via thiol/OPA derivatization. We are particularly interested in developing *N,N*-dimethyl-l-cysteine (DiCys) with OPA as a potential strategy for isotope tags in chiral metabolomics. DiCys can be easily synthesized in one step from l-cysteine by reductive amination (dimethyl labeling), using formaldehyde (CH_2_O) and sodium cyanoborohydride (NaBH_3_CN). The fact that CD_2_O, ^13^CH_2_O, ^13^CD_2_O and NaBD_3_CN are commercially available at relatively low costs means that +2, +4, +6, and +8 Da tags can be easily generated via dimethyl labeling [35,36]. Moreover, ^13^C- and ^15^N-cysteines are also commercially available, which means that up to 10-plex isotope labeling (+0 – +9 Da) is feasible (Appendix A).

Due to the potential of DiCys/OPA as a versatile isotope-labeling method, this study sought to understand its performance in standard RP-LC-MS analyses of chiral metabolites. DiCys was evaluated against five commonly used chiral thiols: NAC, NAP, IBLC, NBC, and BTCC. The reaction mechanism of DiCys/OPA with amines is shown in Figure 1a, and the chemical structures of the other thiols are shown in Figure 1b. They were compared based on their fluorescence intensity, separation performance, stability, and ionization efficiency for amino acid enantiomers. DiCys/OPA was also compared against Marfey’s reagents, which are commonly used for resolving chiral amino acids. Finally, we compared DiCys against IBLC in identifying d-amino acids in aqueous extracts of rice. Our data suggest that DiCys/OPA is an excellent derivatization method to resolve chiral amines in RP-LC-MS metabolomics profiling.

## 2. Results and Discussion 

### 2.1. Stability and Fluorescence of DiCys Derivatives

Some of the most abundant amine-containing metabolites in biological samples are amino acids. l and d amino acid pairs are also among the most important enantiomeric metabolites in terms of biological functions. The charged carboxylate group makes it somewhat challenging to resolve all 20 proteinogenic amino acids by RP-HPLC. Therefore, we chose amino acids as model metabolites to study DiCys/OPA derivatization. 

One of the reported disadvantages of OPA/thiol derivatization is the instability of the product [24,25]. Here, we evaluated the stability of OPA adducts with DiCys and five additional chiral thiols—NAC, NAP, IBLC, NBC, and BTCC. We monitored the fluorescence intensities of OPA/thiol-derivatized amino acids at 4 °C for 60 min (Appendix A), and there was no visible sign of fluorophore breakdown, consistent with previous reports [37]. This should therefore be stable enough for routine LC-MS workflows. 

We also quantified the fluorescence intensities of five l-amino acids derivatized with six chiral thiols (Figure 2a) after HPLC separation. Our results indicated that IBLC, NAC, and DiCys derivatives produced stronger fluorescence. In contrast, the NAP and NBC derivatives exhibited very low fluorescence intensities.

### 2.2. Separation of Enantiomers 

When Chernobrrovkin et al. compared NAC, NAP, IBLC, and NMC as chiral thiols for OPA derivatization [33], they found that NAC and NMC provided better chiral resolution than NAP and IBLC. However, the resolution factors may depend on the column; mobile phase composition; flow rate; and gradient [38,39]. We previously found that optimal elution condition for OPA adducts was around pH 8 instead of the typical acidic conditions [40]. Therefore, we conducted RP-LC separation at pH 8 to resolve five enantiomer pairs (Glu, Ser, Ala, Tyr, and Phe) (Figure 3). The best resolution was obtained with IBLC and DiCys, and the worst was with BTCC (Appendix A). Quantitative conversion to derivatized products for both enantiomers and the lack of racemization were confirmed by MS detection.

### 2.3. Ionization Efficiency and MS/MS Properties

Amino acids exhibit low ionization efficiencies in ESI-MS experiments, and OPA derivatization may bring significant enhancements [40]. As shown in Figure 2b, all six thiol adducts have shown 25–100-fold higher ionization efficiencies over non-derivatized amino acids, making them generally useful for ESI-MS detection. In MS analyses, it is easy to identify derivatized amino acid enantiomers in the mass chromatogram based on double-peak detection via selected ion monitoring. To fragment OPA adducts requires relatively high collision energies: around 20 V [40]. The fragmentation patterns of DiCys/OPA adducts with seven amino acids are shown in Appendix A, with a neutral loss of the thiol group in all cases.

### 2.4. Comparing DiCys/OPA against Marfey’s Reagents

For the enantiomeric separation of amino acids, Marfey’s reagent has been used widely [41,42,43,44]. This has led to the development of several Marfey variants, including: 1-fluoro-2-4-dinitrophenyl-5-l-alanine amide (FDAA); 1-fluoro-2-4-dinitrophenyl-5-l-Valina amide (FDVA); and the corresponding Phe, Ile, and Leu versions [44]. Here, we compared the most commonly used Marfey’s variants, FDAA and FDVA, to the performance of DiCys/OPA. DiCys was the best of the three regarding chiral amino acid separation (Appendix A). DiCys/OPA derivatives also have the advantage of being fluorogenic, while Marfey’s derivatives are non-fluorescent. Therefore, we conclude that DiCys/OPA is highly suitable for resolving chiral analytes, better than popular methods such as NAC/OPA and Marfey’s reagents.

### 2.5. Enantiomer Identification in Rice Water with DiCys/OPA

To test the usefulness of DiCys/OPA, we analyzed the aqueous extracts of edible rice, otherwise known as rice water. Rice water is the starchy water that remains after soaking or cooking rice, containing vitamins, amino acids, and minerals. It has been used traditionally in the treatment of skin and hair in Asian countries [45,46,47]. Little is known about the composition of amino acid enantiomers in rice water. Therefore, we separately applied DiCys/OPA and IBLC/OPA derivatization to rice water samples. Their RP-HPLC chromatograms are shown in Figure 4, and we observed almost twice as many visible fluorescent peaks with DiCys compared to IBLC. It shows that DiCys is suitable for separating a wide range of naturally occurring amine metabolites.

By MS and MS/MS detection, we could identify all twenty proteinogenic l-amino acids and seven d-amino acids (Ala, Arg, Lys, Phe, Ser, Tyr, and Val) in rice water samples using either DiCys or IBLC. Figure 5 shows the integrated ion intensities of individual amino acids. The retention time, precursor ion, and product ion information are listed in Appendix A. The ratios between d/l amino acids are shown in Table 1. Interestingly, the highest d/l ratios were found for the two positively charged amino acids, Arg and Lys. Their physiological roles and gustatory effects remain undetermined.

## 3. Materials and Methods

### 3.1. Reagents

l and d amino acids (Glu, Ser, Ala, Tyr, Phe), l-Cys, OPA, ammonium bicarbonate, perchloric acid (ACS reagent, 70%), formaldehyde (37% *w*/*w*), dichloromethane, ninhydrin, fluorescamine, Ellman’s reagent (DTNB), NAC, NAP, IBLC, NBC, and BTCC were purchased from Sigma-Aldrich (St. Louis, MO, USA). Methanol and acetonitrile (ACN) were purchased from Baker (Radnor, PA, USA). Boric acid and sodium tetraborate were purchased from Acros (Geel, Belgium). Sodium cyanoborohydride was purchased from Fluka (Buchs, Switzerland). FDAA and FDVA were purchased from Thermo Fisher (Waltham, MA, USA).

### 3.2. Synthesis of N,N-Dimethyl-l-Cysteine

A total of 100 mg of l-cysteine was dissolved in 10 mL dilute HCl (pH 2.5) and mixed with 8.25 mmol of sodium cyanoborohydride (NaBH_3_CN) for 10 min at 4 °C. Then, 8.25 mmol of formaldehyde (37% *w*/*w*) was added, stirred for 30 min, and the reaction was monitored by ninhydrin staining on thin-layer chromatography plates. The DiCys product was purified via silica-gel column chromatography using MeOH/CH_2_Cl_2_. DiCys fraction was acidified to pH 2.5 by adding 0.1 N HCl and evaporated to dryness at 60 °C. DiCys powder was dissolved in deionized water and quantified using the Ellman assay. The reaction yield was 87%. HRMS (ESI/Q-TOF) *m/z*: M = C_5_H_11_NO_2_S, calculated for [M + H]^+^ = 150.0583, found 150.0589.

### 3.3. Rice Water Preparation 

Sushi rice samples were purchased from a local grocery store in Taiwan. In total, 50 g of the dried rice was placed in 50 mL of deionized water. After shaking for 30 min, the solution was passed through filter paper. The rice water was lyophilized and dissolved in 250 μL of 0.01% perchloric acid and filtered twice through 0.22 μm nylon filters. Finally, we quantified total amines using a fluorescamine assay [48].

### 3.4. Derivatization Reactions

The following reagents were prepared freshly before use: l, d-glutamic acid; l, d-serine; l, d-alanine; l, d-arginine; d-valine; l, d-tyrosine; l, d-phenylalanine; and l, d-lysine. These reagents were used as amino acid standards and dissolved in 0.01% perchloric acid. The thiols (DiCys, NAC, NAP, IBLC, NBC, and BTCC) were dissolved in methanol to 150 mM. The OPA solution (50 mM) was prepared by dissolving 1.5 mg OPA in a mixture of 20 μL MeOH and 180 μL of 1 M borate buffer (pH 10.7); then, 5.03 mg of FDAA was dissolved in 500 μL ACN (37 mM), and 5.55 mg of FDVA in 500 μL ACN (37 mM). 

After this, 20 µL of 50 mM OPA, 20 μL of 1 M borate buffer (pH 10.7), and 20 µL of 150 mM thiol were combined. We then added either 20 μL of 2.5 mM amino acid solution or rice water sample, and the mixture was vortexed and incubated at 4 °C for 2 min under dark conditions. The solution was diluted to a final volume of 200 µL with 50% ACN, and 20 µL of the mixture was injected into the HPLC. 

A total of 20 µL of 37 mM Marfey’s reagent (FDAA or FDVA) was mixed with 20 μL of 2.5 mM amino acid solution, 8 μL of 1 M NaHCO_3_ (pH 8.0), and 31.5 μL of acetone. The mixture was incubated at 40 °C for 1 h, quenched by adding 6 μL of 2 M HCl, before 20 µL of the mixture was injected into the HPLC.

### 3.5. LC-MS Analysis

The Agilent 1260 HPLC system (Santa Clara, CA, USA) was equipped with an autosampler, a quaternary pump, a column oven, a UV-Vis absorbance detector, and a fluorescence detector. The Hydrosphere C18 column (250 × 4.6 mm, 5 μm bead diameter) used for separation was acquired from YMC (Kyoto, Japan). The aqueous mobile phase (A) consisted of 2 mM ammonium bicarbonate (pH 8.0), whilst mobile phase B contained 7% MeOH in ACN. Elution was performed at a flow rate of 1 mL/min at 40 °C using the following gradient program: 0–5 min, 10%; 5–10 min, 10–12%; 10–20 min, 12–22%; 20–30 min, 22–38%; 30–40 min, 38–60%; 40–47 min, 60–83%; 47–50 min, 83–100%; 50–54 min, 100%; 54–57 min, 100–10%; 57–60 min, 10%. The HPLC was connected to Bruker micrOTOF-QII (Bremen, Germany) operated in positive mode. Full MS spectra were recorded from (*m*/*z*) 100 to 600. ESI source parameters were nebulizer gas (nitrogen) at 0.3 bar, drying gas (nitrogen) at 4 L/min, and 180 °C.

## 4. Conclusions

We systematically evaluated the suitability of six chiral thiols (DiCys, NAC, NAP, IBLC, NBC, and BTCC) for OPA-assisted separation of amino acid enantiomers. The best separation efficiencies in RP-HPLC were found with DiCys and IBLC. For fluorescence detection, IBLC, NAC, and DiCys gave stronger signals while NAP only gave very weak signals. All six reagents enhanced ionization efficiencies by 25–100 fold, useful for MS detection. Previously, IBLC has been a popular reagent for resolving chiral amino acids [30,49], and our data supported its usefulness. More importantly, our study was the first to introduce DiCys/OPA for enantiomeric separation, and its performance was comparable to IBLC in our tests. DiCys also outperformed Marfey’s reagents FDAA and FDVA, which were specially developed for chiral separation purposes. We conclude that DiCys is a highly versatile reagent for resolving enantiomeric amines in chiral metabolomics experiments.

The greatest advantage of DiCys is its potential for multiplex heavy-isotope labeling. Using well-established chemistries [35], heavy isotope versions from +1 Da to +9 Da may be easily synthesized in one step using commercially available reagents. This may provide 10-plex labeling at an affordable cost for high-throughput metabolomics experiments. Combined with fluorogenic detection and excellent chiral separation, DiCys is one of the most versatile amine derivatization reagents currently available.

In real-world metabolomics profiling of rice water, DiCys provided better separation of amine metabolites than IBLC. Both allowed us to detect twenty proteinogenic l-amino acids and identify seven d-amino acids—Ala, Arg, Lys, Phe, Ser, Tyr, and Val. These d-amino acids are primarily associated with sweetness for humans [50,51], suggesting that d-amino acids may be important for the gustatory taste. Moreover, d-amino acids synthesized by gut microbiomes may affect our immune systems [52,53]. How d-amino acids in rice diet may affect our gut microbiome–immune axis will warrant further investigation.

## Figures and Tables

**Figure 1 molecules-26-07416-f001:**
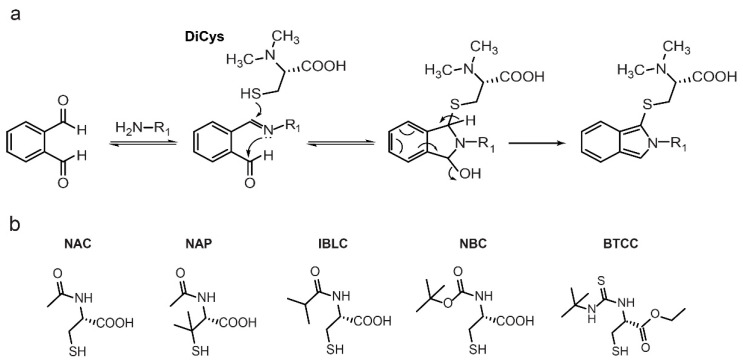
(**a**) Derivatization reaction of amino acids with DiCys/OPA. (**b**) Structures of chiral thiols: NAC, NAP, IBLC, NBC, and BTCC.

**Figure 2 molecules-26-07416-f002:**
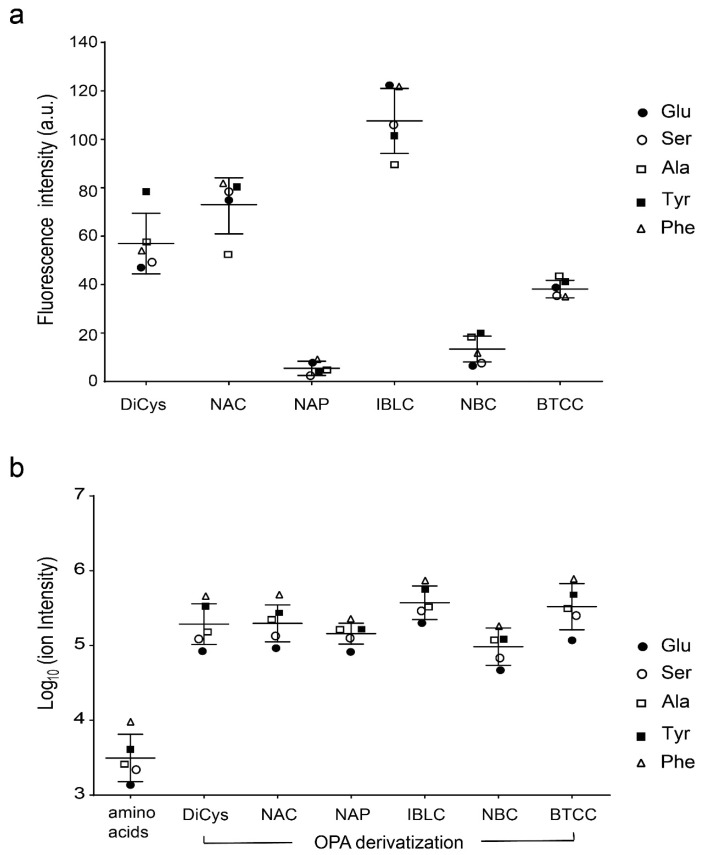
Fluorescence intensities (**a**) and ion intensities (**b**) of the OPA/thiol-derivatized l-amino acids (Glu, Ser, Ala, Tyr, and Phe) at equal concentrations. The center bar represents the mean, and the whiskers represent ±2 standard deviations.

**Figure 3 molecules-26-07416-f003:**
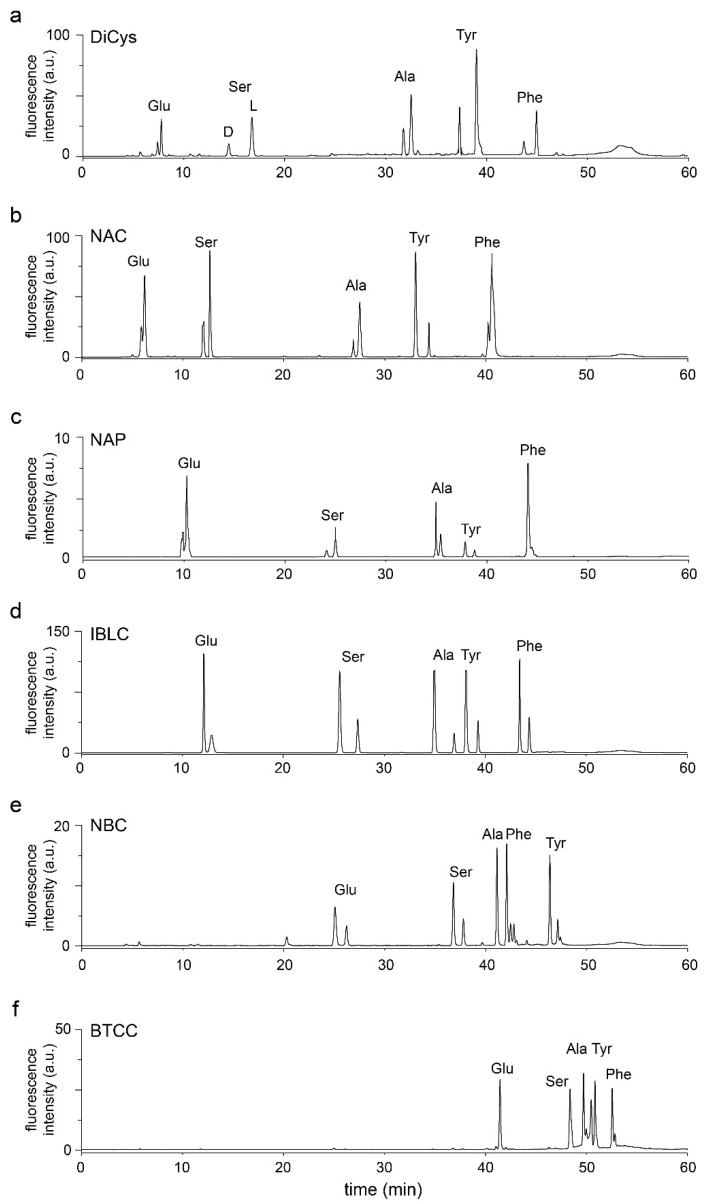
RP-HPLC analysis of amino acid enantiomers derivatized with OPA/thiol. The adducts of DiCys (**a**), NAC (**b**), NAP (**c**), IBLC (**d**), NBC (**e**), and BTCC (**f**) are detected by 340 nm excitation/450 nm emission. The ratio between l:d amino acids is 3:1.

**Figure 4 molecules-26-07416-f004:**
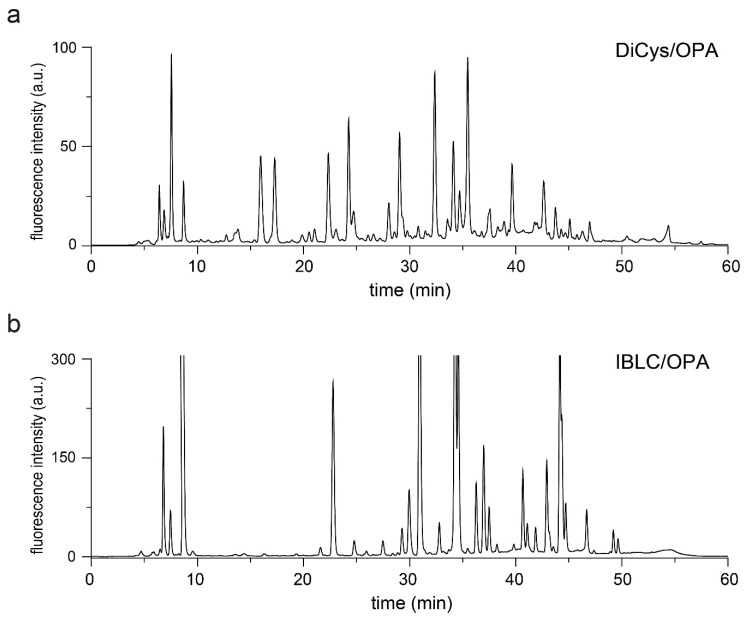
RP-HPLC analysis of free amines in rice water derivatized by DiCys/OPA (**a**) and IBLC/OPA (**b**), detected by 340 nm excitation/450 nm emission.

**Figure 5 molecules-26-07416-f005:**
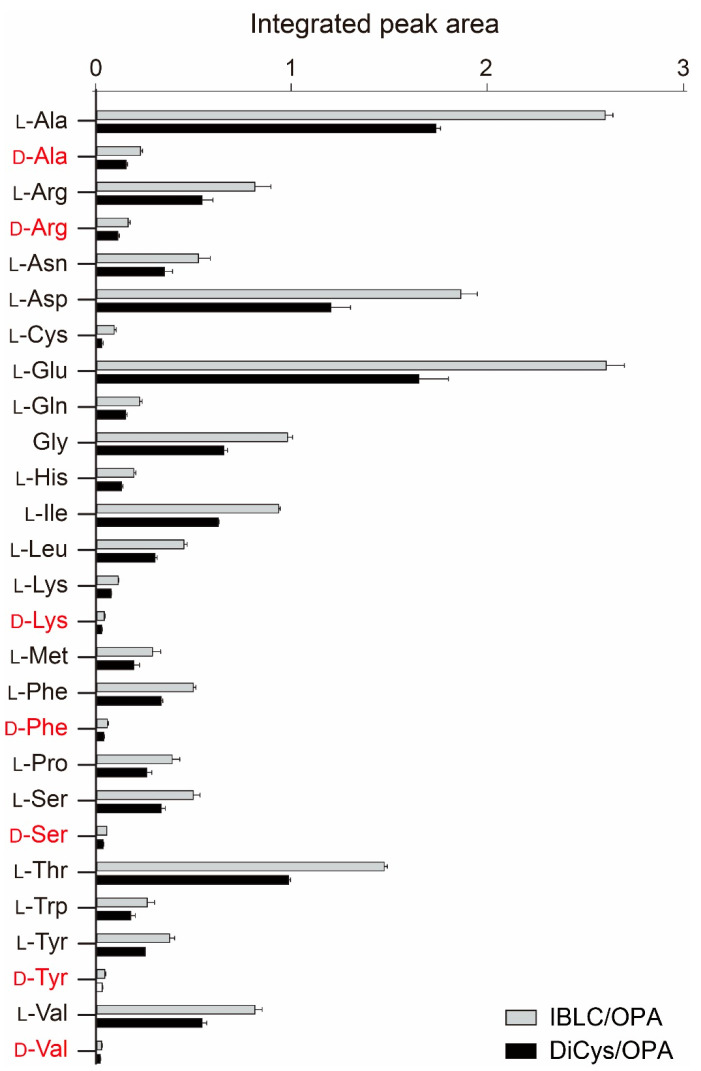
The amino acid contents of rice water measured by IBLC/OPA and DiCys/OPA derivatization. Bars indicate the integrated peak area of ion intensities from RP-LC-MS analyses. Error bars correspond to the standard error from three independent replicate experiments.

**Table 1 molecules-26-07416-t001:** The ratios of d-amino acid to l-amino acid in rice water.

Amino Acids	d/l Ratio
Ala	0.09
Arg	0.31
Lys	0.30
Phe	0.09
Ser	0.10
Tyr	0.12
Val	0.03

## Data Availability

Not applicable.

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
