# Peer review of "Dimethylcysteine (DiCys)/o-Phthalaldehyde Derivatization for Chiral Metabolite Analyses: Cross-Comparison of Six Chiral Thiols"

_molecules, 2021, doi:10.3390/molecules26247416_

Round 1

Reviewer 1 Report

The article entitled “Dimethylcysteine (DiCys)/o-phthalaldehyde derivatization for chiral metabolomics: cross-comparison of six chiral thiols” by Lkhagwa etal., delas with the development of an improved method for separation of chiral amino acids using dimethylcysteine assisted OPA derivatization technique. The authors successfully applied their technique to profile D/L amino acids in aqueous extracts of rice. The study is very interesting and need some minor modifications before publishing in molecules. I have the below comments.

  1. Section 3.3 : “All six thiol adducts have shown 25-100 fold higher ionization efficiencies over non-derivatized amino acids, making them generally useful for ESI-MS detection” does authors have any data to prove this statement?
  2. Although authors provided MS and MS/MS ions it would be great if they provide the original MS/MS spectra all the AA-derivatives (either D or L) as supporting information. Currently, only alanine is shown.
  3. Do authors detect any side product peaks in the HPLC chromatograms? Its common to observe by-products during OPA derivatization. The fluorescence intensity units should be denoted in “arbitrary unit”
  4. The references cited are old (ex: Ref.28, 29, 35, 36, 42). Authors must refer recent articles and cite them accordingly. Author may refer the latest articles related to RP-LC-MS/MS (Anal Sci. 2020 Jul 10;36(7):821-828, Metabolites 202010(10), 398) and OPA-derivatization i.e., successfully applied in detection and separation of chiral compounds ( Lett. 2016, 18, 10, 2327–2330, ACS Omega 2018, 3, 1, 753–759).

Page 2: “method to revolve chiral amines” should be method to resolve

Section 2.2: The yield of the reaction is missing. How did the authors confirm the demethylation? Either HRMS or NMR data should be provided

Section 2.3: What is the recovery percentage of amino acid extraction from rice?

Table S3: m/z should be written in italics throughout the manuscript.

Reviewer 2 Report

The manuscript describes a new derivatizating agent to separate chiral amines (mainly amino acids) by LC-MS.

I consider the manuscript shows some wrong concepts, such as metabolomics, as well as some confusing data, which does not make sense to be presented. Please, find some specific points below.

Title:

In my opinion the title and the abstract should clearly show that this is targeted metabolomics, since the method is applied to amino acids analyses. Besides, the authors did not perform metabolomics analyses, since there was no comparison of metabolites profiles between groups of samples submitted to different alterations.

Abstract and all long the text:

The manuscript proposes LC-MS analyses. Why is fluorescence intensity an important parameter?

What is the importance of quantifying amino acids enantiomers in rice?

Introduction:

“The fact that CD2O, 13CH2O, and NaBD3CN are commercially available at relatively low cost means that +2, +4, +6, and +8 Da tags can be easily generated via dimethyl labeling [33, 34]. Moreover, 13C- and 15N-cysteines are also commercially available, which means that up to 10-plex isotope labeling (+0 ~ +9 Da) is feasible”: authors should make clearer how these tags reach this mass increments.

“Finally, we compared DiCys against IBLC for identifying D-amino acids in aqueous extracts of rice”: what about NAC? Why was DiCys not compared to it as well, if it showed similar stability and fluorescence intensity?

Figure 1: does this reaction occurs concomitantly? I mean: OPA DiCys and amino acids? The figure should show each step of the reaction and insert arrows (reaction mechanism) to the reader understand what groups / atoms are involved

Material and methods:

2.2: the reaction should be presented as a Figure

Normality and equivalents are not commonly used anymore. Replace by molarity and the respective amounts.

“Five equivalents (37 mM) of Marfey’s reagent (FDAA or FDVA) (20 μL) were added to 20 μL of amino acid standards (2.5 mM), 8 μL of 1 M NaHCO3 (pH 8.0), and 31.5 μL of acetone”: the reaction should be presented as a Figure.

Results:

Figure 2: I can not figure out which is each amino acid fluorescence or ion intensity. The figure should be elaborated again.

“resolve five enantiomer pairs (Glu, Ser, Ala, Tyr, and Phe) (Fig. 3). The best resolution was obtained with DiCys and IBLC, and the worst was with BTCC (Supplementary Table. S1). Quantitative conversion to derivatized products for both enantiomers and the lack of racemization were confirmed by MS detection.” Table S1 does not show information for BTCC

Figure 3: indicate on each peak which is the D and which is the L enantiomer

“As shown in Fig. 2b, IBLC, BTCC, and DiCys adducts have higher ionization efficiencies compared to other thiols”: I don´t see this conclusion in Figure 2 B, since the standard deviations do not allow differentiation of ionization efficiency.

“DiCys was equally effective as FDVA and clearly outperformed FDAA. DiCys/OPA derivatives have the advantage of being fluorogenic but Marfey’s derivatives are non-fluorescent”: what were the parameters to achieve this conclusion? what is the advantage of being fluorescent if the analysis is performed by LC-MS?

Figure 5: was it obtained from LC-MS or LC-fluorescence? are these sample preparation replicates or injections of the same sample? It must be different samples injected once each one.

Supplementary materials:

Table S1: indicate on the table foot why there are blank information on some lines

Figure S3-A: where is Val peak?

Round 2

Reviewer 2 Report

There are still modifications that must be performed. For instance: the manuscript does not report a metabolomic study (as I pointed out in the first revision), since it does not compare groups of samples (where one has been altered and the other one is a control). What the authors have done is the determination of amino acids in the samples. In the experimental session, the authors have replaced normality by molarity, but still use the term "equivalents"

Author Response

There are still modifications that must be performed. For instance: the manuscript does not report a metabolomic study (as I pointed out in the first revision), since it does not compare groups of samples (where one has been altered and the other one is a control). What the authors have done is the determination of amino acids in the samples.

>> We have removed "metabolomics" from our title and abstract.

New title: Dimethylcysteine (DiCys)/o-phthalaldehyde derivatization for chiral metabolite analyses: cross-comparison of six chiral thiols

Last sentence of abstract: We conclude that DiCys/OPA is an excellent amine-derivatization method for enantiomeric metabolite detection in LC-MS analyses.

In the experimental session, the authors have replaced normality by molarity, but still use the term "equivalents"

>> We have removed "equivalent" in section 2.2, see below:

1 g of L-cysteine was dissolved in 100 mL dilute HCl (pH 2.5), and mixed with 82.5 mmol of sodium cyanoborohydride (NaBH3CN) for 10 min at 4 °C. 82.5 mmol of formaldehyde (37% w/w) was added....